# StatTexNet: Evaluating the Importance of Statistical Parameters for Pyramid-Based Texture and Peripheral Vision Models

**C. Koevesdi**                                                                              koevesdc@mit.edu
**V. DuTell**                                                                                    vasha@mit.edu
**A. Harrington**                                                                              annekh@mit.edu
**M. Hamilton**                                                                                markth@mit.edu
**W. T. Freeman**                                                                                billf@mit.edu
**R. Rosenholtz**                                                                               rruth@mit.edu
*MIT CSAIL, Brain and Cognitive Sciences*

## Abstract

Peripheral vision plays an important role in human vision, directing where and when to make saccades. Although human behavior in the periphery is well-predicted by pyramid-based texture models, these approaches rely on hand-picked image statistics that are still insufficient to capture a wide variety of textures. To develop a more principled approach to statistic selection for texture-based models of peripheral vision, we develop a self-supervised machine learning model to determine what set of statistics are most important for representing texture. Our model, which we call StatTexNet, uses contrastive learning to take a large set of statistics and compress them to a smaller set that best represents texture families. We validate our method using depleted texture images where the constituent statistics are already known. We then use StatTexNet to determine the most and least important statistics for natural (non-depleted) texture images using weight interpretability metrics, finding these to be consistent with previous psychophysical studies. Finally, we demonstrate that textures are most effectively synthesized with the statistics identified as important; we see noticeable deterioration when excluding the most important statistics, but minimal effects when excluding least important. Overall, we develop a machine learning method of selecting statistics that can be used to create better peripheral vision models. With these better models, we can more effectively understand the effects of peripheral vision in human gaze.

**Keywords:** peripheral vision, texture synthesis, multi-scale pyramid, statistic selection, contrastive learning

## 1. Introduction

A key source of information in human gaze comes from peripheral vision. While it is often thought of as an adaptation to capacity limits of the human visual system, peripheral vision also drives human performance on many visual tasks – including search, scene perception, and object detection (Ehinger and Rosenholtz, 2016). With respect to gaze specifically, peripheral vision plays a role in saccadic planning by helping humans determine where to look next (Schütz et al., 2011).

Given its importance in understanding human gaze patterns, numerous attempts have been made to model peripheral vision. Multi-scale-pyramid-based models are the current

state of the art. These models account for both the loss of photoreceptor density and the summarization of information thought to occur in brain areas V2 and V3. Models such as these treat peripheral vision as a texture-like representation and have a long history in human vision. They have been used in not only peripheral vision, but also in texture models more generally (Portilla and Simoncelli, 2000). To simulate peripheral vision, these models utilize overlapping pooling regions that encircle the fovea and increase in size with eccentricity. While some models utilize machine learning techniques like style transfer (Wallis et al., 2017; Deza et al., 2017) to summarize information, the majority of these models calculate summary statistics for each pooling region, which are calculated on the output of multi-scale pyramids.

One challenge of pyramid-based models of peripheral vision is in determining which statistics are calculated in each pooling region. Although most pyramid-based texture models used to study peripheral vision have been validated through human behavioral studies, they still utilize statistic sets that are historically driven, vary study-to-study from previous literature, and are consistently insufficient to capture the wide variety of possible textures Brown et al. (2021).

The problem of selecting which statistics are necessary and sufficient to represent the variety of textures perceived in peripheral vision is critical for the goal of building better models of human gaze. While testing every single texture by hand or with a human-in-the-loop is not feasible, we leverage self-supervised approaches in machine learning to address the problem of statistic selection in peripheral vision models. In this work, we develop a constrastive learning model, StatTexNet, to explore the relative importance of pyramid-based statistics for representing peripheral vision. To validate our machine learning approach to statistic selection, we test our framework on a set of depleted texture images with known statistics. We demonstrate that StatTexNet selects the known most important statistics in these depleted textures. We then apply our model to full texture images and use weight interpretability metrics to determine what are the most important statistics to represent texture families. Finally, we synthesize textures using statistics selected by our method.

By building a machine-learning-driven approach to statistic selection, our work automates the evaluation of statistics used by texture-based peripheral vision models. With a better method of understanding and evaluating peripheral vision models, we can build a more complete understanding of human gaze.

## 2. Previous Work

Peripheral vision represents the majority of the visual field, and both critically limits and enables human performance at a variety of tasks (Rosenholtz, 2016). This includes gaze behavior where information from both the fovea and the periphery are integrated to inform saccades (Stewart et al., 2020).

Some of the best performing models of peripheral vision use a multi-scale pyramid approach. Most pyramid-based peripheral vision models are based on work from the texture modeling world. Early work in this area included (Julesz, 1962), who first explored different textures that could be represented as the same N-th order pixel statistics. Large improvements were seen with a move from pixel-based to multi-scale pyramid based sta-

tistical representations (Simoncelli and Freeman, 1995). The steerable pyramid has since been widely used in vision modeling as its filters resemble those found in the mammalian early visual system (Turner, 1986; Malik and Perona, 1990), which break down an input image into distinct spatial frequency and orientation bands. Using the steerable pyramid, Heeger and Bergen (Heeger and Bergen, 1995) proposed a statistics set calculated on this pyramid decomposition, alongside a histogram-matching procedure that enabled good texture synthesis. This was refined further by (Portilla and Simoncelli, 2000), which included pixel, autocorrelation, and magnitude statistics.

When these texture models were first applied to peripheral vision (Rosenholtz et al., 2012; Freeman and Simoncelli, 2011), they utilized a similar texture set to (Portilla and Simoncelli, 2000). Statistics were modified from this set by being hand-chosen and tested for necessity and sufficiency through trial and error on a limited test set of textures. More recent work has modified these statistics slightly, tested them on a wider variety of conditions and textures, and made code more flexible and efficient (Brown et al., 2021; Wallis et al., 2017).

Behavioral evidence supports the statistics set used by these state-of-the-art peripheral vision models. These models are often used to create mongrels, also known as metamers, which are visual stimuli that match another in representational space, but can differ significantly in pixel space. When viewed foveally, the pixel-differences are obvious, but when viewed peripherally, they are indistinguishable. Mongrels have been shown through careful psychophysical experimentation to reproduce the same capabilities and limitations of human peripheral vision including crowding (Balas et al., 2009) and scene perception (Ehinger and Rosenholtz, 2016). In addition, the scaling parameters for pooling regions needed to create metamers/mongrels mirror those of neuron receptive fields in non-human primates (Freeman and Simoncelli, 2011).

Despite the success of these models, it is clear that the current state-of-the-art statistic set is insufficient. A faithful model of human peripheral vision should work regardless of input type. However, investigations into the effect of different texture families have revealed that for current models, textures with certain properties are more faithfully represented, while metamers/mongrels of other texture types consistently fail (Brown et al., 2021; Broderick et al., 2023). These problems occur despite modifications to optimization strategy, hyperparmeters, and seed.

Some efforts have worked to eliminate the need to choose specific statistics altogether. Mongrels have been successfully created by taking inspiration from style transfer (Gatys et al., 2016), utilizing the entire gram matrix as the statistical representation to create metameric images (Deza et al., 2017; Wallis et al., 2017). While this removes the need for hand-picking statistics, this represents a huge matrix that is likely over-parameterized, and removes any potential compression advantage. Another example is the work from (Serre et al., 2007), which simply takes the maximum output of each pooled area. Although the field has made significant progress toward improving the statistic component of peripheral vision models, it is clear that a more principled approach to selecting the statistics is needed.

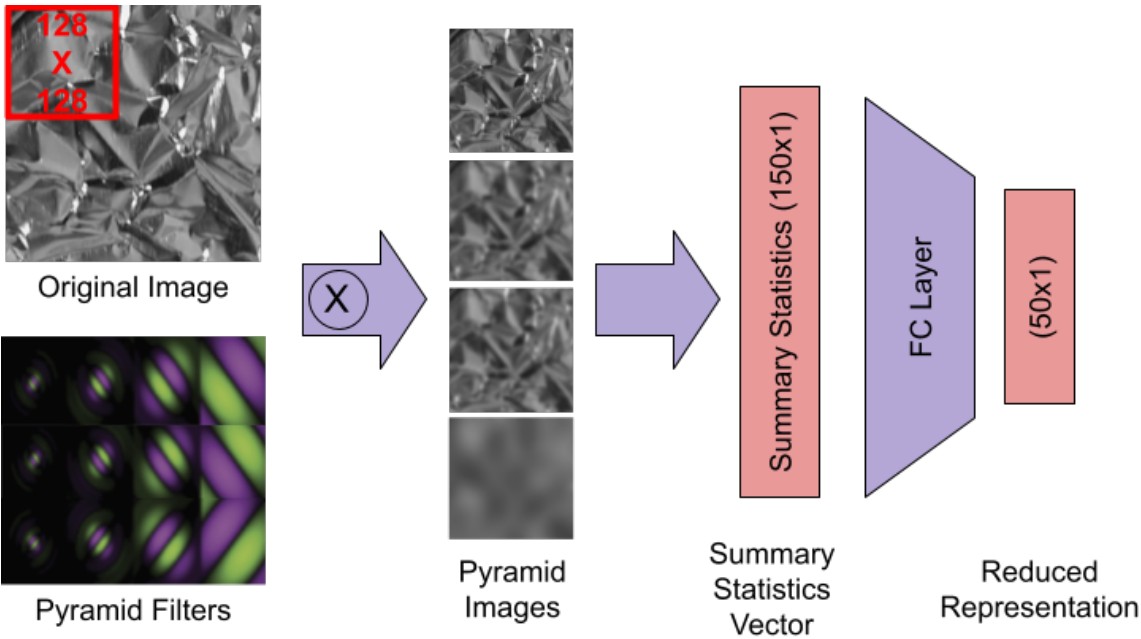

Figure 1: Our model compresses the representation of a texture model.

## 3. Modeling Textures Through Statistics

In order to build a better method of selecting the most important statistics for texture-based peripheral vision models, we devise a contrastive learning model, StatTexNet, to take a large set of statistics and compress it to a smaller set. In our model, we take 5-crops (4 corners and center) from a texture image dataset, and calculate their summary statistic representation using the GPU-optimized code from (Brown et al., 2021) (Figure 1). This consists of convolution of each 128x128 pixel crop with a steerable pyramid filter bank, and the calculation of 150 summary statistics from these pyramid images. We then use a single fully connected layer to compress this statistical representation, which we train through contrastive learning. The input space is thus 150. For the output latent space, we choose a dimensionality of 50, as it provides the most effective clustering in our experiments. While we use the statistics set from the (Brown et al., 2021) model as a baseline, we note that this is a similar statistics set to other popular models (Portilla and Simoncelli, 2000; Freeman and Simoncelli, 2011; Rosenholtz et al., 2012), with some statistics removed for computational savings, simplicity, and based on empirical findings, as well as the inclusion of an additional statistic set, 'end-stopped'.

## 4. Summary Statistics Sets

StatTexNet starts with an initial set of summary statistics which are are split into two groups: *first-order* and more complex *second-order and higher* statistics.

Following (Brown et al., 2021), we utilize the following statistics:

**First-order statistics:**

- From the raw input image pixels, the first four moments — mean, variance, skewness, and kurtosis — of the grayscale histogram.

- The variance of both the high- and low-pass bands, with skewness and kurtosis also computed for the latter.

- For the non-oriented lowpass bands, the variance, skew and kurtosis are computed.

- For each bandpass filter output, the magnitude-mean and variance are derived.

**Second- or higher-order statistics:**

- Magnitude-correlations between bandpass filters. This involves the correlations between all orientations at the same scale in the steerable pyramid, but also correlations between neighboring scales at the same orientations.

- The same correlations are also computed for the phase images.

- Finally, unique to Brown et al is the *End-Stopped* statistic. This statistic is based on end-stopped neurons or hypercomplex cells in visual cortex (Hubel and Wiesel, 1959), and differentiates between segmented and continuous lines. Specifically, each edge magnitude component image is subtracted from a slightly shifted version of itself, following the expected edge direction. The resulting difference is then squared.

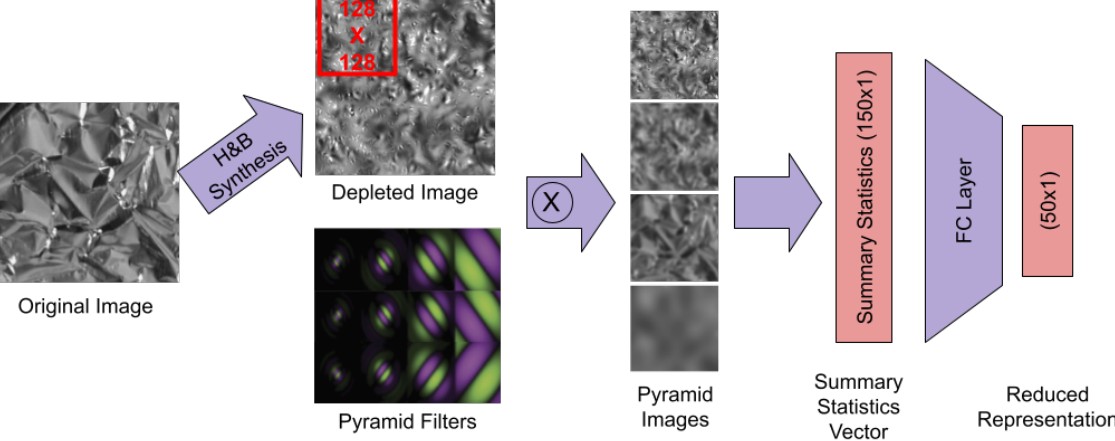

Figure 2: We generate depleted textures created with a known set of statistics, feeding these controlled images to our model, and perform the same procedure as in Figure 1.

Natural images are essentially unbounded by the set of statistics that represent them. However, synthesized images are created using only a set of known statistics. In order to control for the set of statistics present in a given texture image and validate our method of

statistic selection, we create synthesized versions of each texture using the Heeger & Bergen texture model (Heeger and Bergen, 1995). Heeger and Bergen preforms histogram matching on first order statistics *only* and thus its syntheses are only constrained to this subset of statistics. These synthesized textures are *depleted* in that they do not contain the full set of statistics needed to fully describe them. They can therefore be used to validate our method, as a model should not need higher-order statistics to group them. This enables us to test if a network can learn the relative importance of different groups of statistics from different datasets. To do this, we then follow the same pipeline as in Figure 1, with these depleted images (Figure 2).

## 5. Datasets

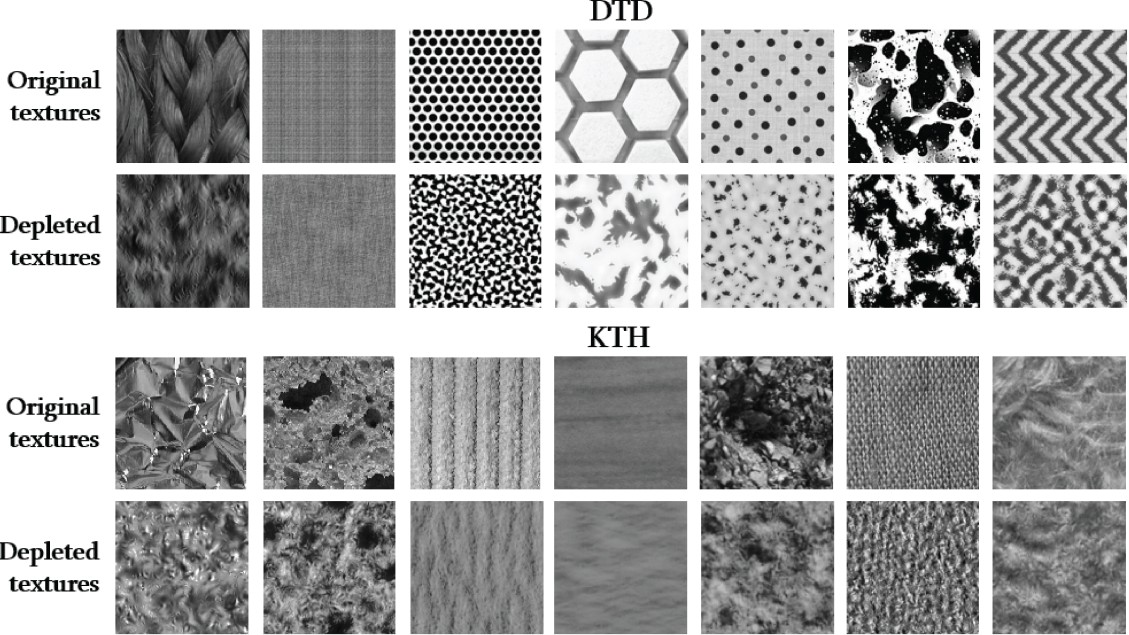

Figure 3: Dataset visualization through sample textures. The top row indicates the original texture and the bottom row shows the synthesized texture through the Heeger and Bergen procedure.

In this study, we utilized two primary datasets: the Describable Textures Dataset (DTD) (Cimpoi et al., 2014) and the KTH-TIPS2-b (KTH) dataset (Mallikarjuna et al., 2006) which we use for validation. The DTD captures a wide array of textures found in natural settings and is a collection of 5,640 images spanning 47 distinct texture categories. These images were primarily sourced from platforms like Flickr and Google Search. The KTH dataset contains 4,752 images representing 11 different materials that were acquired through imaging 4 different samples for each material, each under varying pose, illumination and scale. Due to the way it was collected, DTD has more intra-class variation than KTH.

We transformed all RGB images from these datasets into grayscale. We then applied the Heeger and Bergen texture synthesis procedure (Heeger and Bergen, 1995) to these grayscale images. The Heeger and Bergen approach is to iteratively modify a gaussian white noise image so that the pixel distributions in its steerable pyramid representation match that of the reference texture. This is done through histogram matching. When provided an input image, histogram matching aims to adjust the image's grayscale pixel value distribution so that it aligns with the histogram of a reference image. Thus, histogram matching adjusts the pixel distribution of an image to match that of a reference, ensuring identical first-order statistics, but not guaranteeing similar spatial structures or correlations between images.

Consequently, we have four datasets at our disposal to test our hypotheses: two are the original grayscale sets (DTD and KTH), and the other two are depleted - derived from the Heeger and Bergen synthesis method applied to DTD and KTH. Figure 3 shows some examples of these datasets.

## 6. Training

### 6.1. Contrastive Learning

Our goal is to reduce the full set of 150 Brown et al. (2021) image statistics to a compressed representation $1/3$ the size, forcing the network to prioritize information from certain textures over others. To do this, we employ constrastive learning (Chen et al., 2020), allowing our network StatTexNet to learn any representation that is useful in discriminating textures. Contrastive learning works by ensuring that similar pairs, such as crops from the same image, are drawn close together in representation space, while distinct pairs are pushed apart based on a specified distance measure. For this task, we utilize generalized lifted structured loss (Hermans et al., 2017) with a Euclidian distance. The advantage of this loss is its ability to effectively process the entire training batch, taking into consideration both closely related pairs (positive anchors) and those that are unrelated (negative pairs). In one training step, all pairs are considered (See Appendix Section 12.2).

For our input data, we take a single texture image from one of our datasets and crop it into 5 smaller images. This gives use a set of 5 images that we know come from the same texture, and thus, should be represented by a very similar statistical values. Crops from the same image are treated as positive samples and crops from different texture images are treated as negative in our framework. We train our contrastive learning networks for 200 epochs. (See 12.3 for details on data augmentation). To process our data efficiently and ensure consistent gradient updates, we selected a batch size of 100. Additionally, after evaluating different optimization techniques, we settled on the Adam optimizer due to its adaptive learning rate and proven success in similar tasks. We used a learning rate of 0.0001.

### 6.2. Dropout

One complication of our model is that correlations between different elements of the 150 statistics set could potentially cause the network to ignore certain highly correlated or anti-correlated statistics. We reasoned that, because our statistics sets represent uniform spatial samples of natural images that have inherent regularities (Ruderman, 1997; Simoncelli and Olshausen, 2001), correlation between statistics was highly likely. This could happen when

multiple statistics correlate sufficiently such that the network learns to rely only on one of the correlated statistics, discounting others.

To address, this we first checked for correlations among statistics (Appendix Section 12.5), and found that indeed, the majority of statistics measured show high correlation with at least one other. We counteract this issue by incorporating dropout during training. By incorporating dropout, some features are set to zero temporarily at random during each forward pass. This prevents the model from becoming too reliant on specific features as it forces the model to learn a more even distribution across correlated groups. Thus, this approach mitigates the effects of multi-collinearity. We find that incorporating dropout greatly improves the results in Table 1, compared to training without dropout.

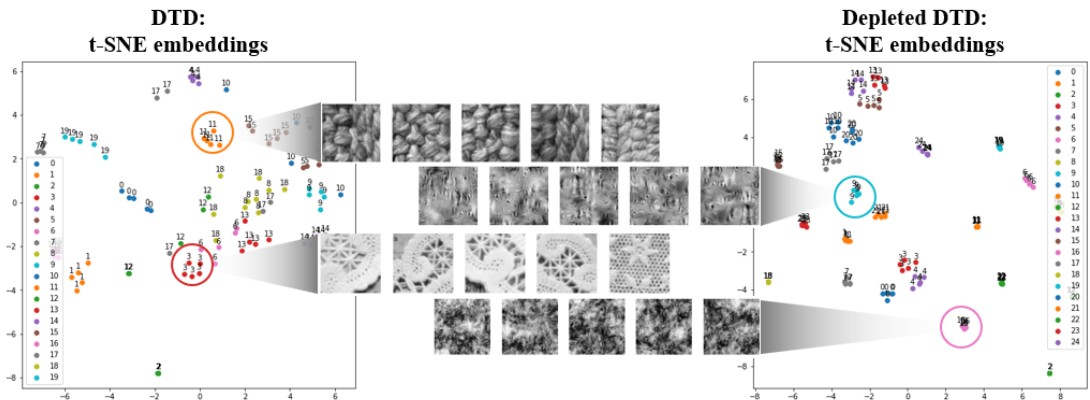

Figure 4: t-SNE visualizations for learned embeddings of the DTD dataset for both original texture images (left), as well as depleted (right). The network learns to cluster textures from both well, while the right plot indicates that depleted images are clustered better.

### 6.3. t-SNE

In addition to seeing a reduction in loss over training, we validated the effectiveness of our contrastive learning approach using t-SNE (Van der Maaten and Hinton, 2008) to visualize the learned latent representation space. To do this, we ran inference on a set of 20 randomly-chosen textures, each with 5 crops, and visualized the 2D embedding of the 50 dimensional space (Figure 4). We find that indeed, crops from the same texture cluster together well in space. This is especially true for the depleted textures synthesized with (Heeger and Bergen, 1995).

## 7. Rankings for Depleted Data

### 7.1. Weight-Based Ordering

After the training process, we do weight-based ordering on StatTexNet to determine the significance of each statistic. We summed up the absolute weight values for each input node

Table 1: Importance metrics for 50 first-order summary statistics averaged over 10 seeds based on two different feature selection methods. For both orderings, all three measures, and both datasets, first-order statistics are ranked higher (are more important) for depleted textures created with these statistics only, than for original textures.

| Ordering | Metric | DTD | | KTH | |
|---|---|---|---|---|---|
| | | Original | Depleted | Original | Depleted |
| **Weight** | **% in Top 15** | 36.00 | 75.33 ✓ | 80.07 | 100.00 ✓ |
| | **Median rank** | 59.45 | 41.60 ✓ | 36.00 | 34.10 ✓ |
| | **Mean rank** | 63.36 | 52.65 ✓ | 46.99 | 44.71 ✓ |
| **Shapley** | **% in Top 15** | 32.67 | 49.33 ✓ | 43.33 | 70.67 ✓ |
| | **Median rank** | 71.20 | 49.45 ✓ | 64.30 | 44.85 ✓ |
| | **Mean rank** | 67.75 | 55.55 ✓ | 62.89 | 50.36 ✓ |

in our model, where each input node corresponds to one of the 150 feature statistics. Because we use scaling in the weight matrix to normalize, a statistic more useful in classification should be weighted more highly by the network. We ordered these weights in descending order, ranking them from the most (low rank) to the least important (high rank).

We followed three metrics to evaluate whether StatTexNet can learn the most important statistics across the different datasets. We hypothesized that the first-order statistics matched by Heeger and Bergen synthesis will play a more important role in the depleted data, than for the original textures. To test this, we calculated the weights for each dataset and then ranked the 150 statistics by their importance. As a first metric, we observed how many of the first-order statistics rank in the top 15 of overall most important statistics. Here, we expect that for the synthesized texture datasets, there will be a higher percentage of very important first-order statistics compared to the original datasets. (We note that raw statistics determine basic image properties, such as brightness - therefore it is expected that even in the original textures, a substantial percentage of the first-order statistics should be among the top 15, though an increase in their prevalance should be expected for the depleted textures.) Furthermore, we assessed both the mean and median importance rank of the 50 first-order statistics in the overall importance ranking. We took the mean value for these three metrics over 10 different seeds and observed consistent results across all of them.

We find that for both datasets, and for all 3 measures (% In Top 15, Median Rank, Mean Rank), relative rankings reflect our expected results. That is, when trained on the depleted dataset, StatTexNet consistently ranked the 50 first-order statistics higher (more important), than when trained on original textures (Table 1, Top). For the KTH dataset, 100% of the top 15 ranked statistics belonged to the first-order statistics for the depleted data (this occurred in all of the 10 separate trainings with different random seeds). This indicates that our framework is sound, and weight-based ordering is able to identify the most and least important statistics for a contrastive learning task.

**7.2. Shapley Value-Based Ordering**

While weight-based ordering using the average absolute value of weights offers strong support of our hypothesis that depleted data would favor first-order statistics more heavily, we sought a more sophisticated mathematical approach to test and validate our findings. Calculating Shapley values (Roth, 1988) is an interpretability method based in game theory enabling the assignment of credit to individual inputs for a given output in a machine learning model. We utilized the SHAP package (Lundberg and Lee, 2017) to calculate Shapley values for each of the 150 statistics, and used these values in place of absolute value of weights to order statistics by importance.

We find that the rankings based on Shapley values also support our hypothesis that depleted texture-trained networks will more heavily rely on the 50 first-order statistics than networks trained on their complete texture counterparts (Table 1, Bottom). Given these results indicating the strong utility of ranking via Shapley values, we chose to utilize this ranking procedure alongside weight in exploring the statistical importance for non-depleted data.

## 8. Statistical Importance for Original Textures

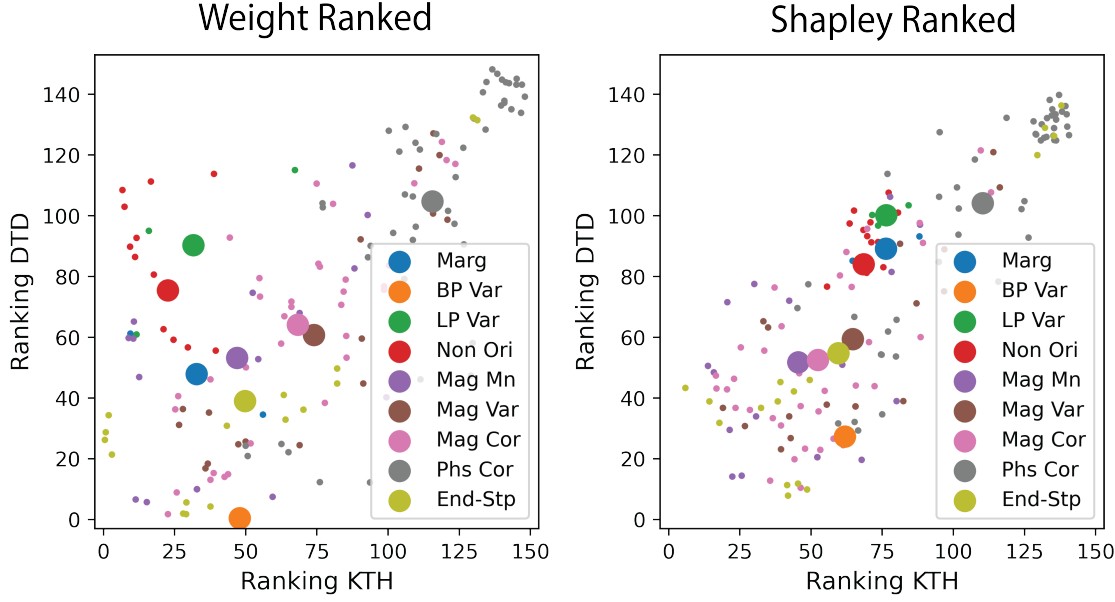

Figure 5: Statistic rankings for two datasets tested. Small points indicate individual statistics, large points indicate group statistic means (circle). Phase-correlation statistics are consistently of low importance, while most other statistics families show heterogeneous performance. Shapley ranking of statistics shows better correlation between datasets tested.

Having validated that our method works in identifying the most and least important statistics for texture representation, we turn to the results on original (non-depleted) textures. First, to understand the relative importance of each statistic, we computed the mean ranking of the nine statistics groups (Figure 5, bar plots in Figure 8), averaged over 10 seeds.

We find that overall, bandpass variance (a single statistic) has high ranking between both datasets and ranking procedures (especially for DTD), indicating that it is important. Magnitude-mean statistics also cluster consistently towards high rankings. Most of the other statistics show a wide distribution of rankings. This is true across datasets, within datatsets, and for both ranking systems. End-stop and magnitude-correlation statistics in particular show highly distributed rankings, appearing as both some of the most and least important statistics.

We find that phase-correlation is consistently ranked far lower than all other statistics classes, with strong rank grouping near the end, indicating that it is a less important statistic overall. Our findings of phase-correlation being less important are consistent with previous psychophysical literature Balas (2006), which found phase-correlation to be unimportant for discriminating textures. Interestingly, only our weight-ranked results are consistent with their findings that marginals are highly important for discrimination.

## 9. Synthesis

One advantage of the texture/peripheral models studied here is the ability to synthesize textures based on a given statistics set. This allows us to visually validate our results. While we emphasize that synthesis results have high variation being both highly seed and texture dependent (Brown et al., 2021; Broderick et al., 2023), we nonetheless include some syntheses here, demonstrating the effect of depleting various statistics.

We show examples of textures with properties found by Brown et al. (2021) to be most and least well-captured by the full texture set. We find that high contrast textures like the lined texture, demonstrate similar performance to baseline (All) when the less-important phase-correlation statistic is removed, but fail completely when the highly-important magnitude-mean statistic is removed. Lower contrast textures, like the painted image, however, show similarly poor synthesis in all cases. The porous texture, lying somewhere in between, has similar synthesis performance to baseline when phase-correlation is removed, and a slightly worse performance when magnitude-mean is removed. Our observations in Figure 8 align with this, highlighting that the magnitude-mean statistics are notably important compared to the phase-correlation statistics. Given that the phase-correlation statistics comprise a greater number of statistics than magnitude-mean, this offers a meaningful point of comparison.

These syntheses support the results uncovered here through our contrastive learning approach. While the 150 statistics of Brown et al. (2021) are not sufficient for all textures, removal of the phase-correlation statistic is often not important, while removal of the magnitude-mean statistic is often noticeable, and sometimes catastrophic.

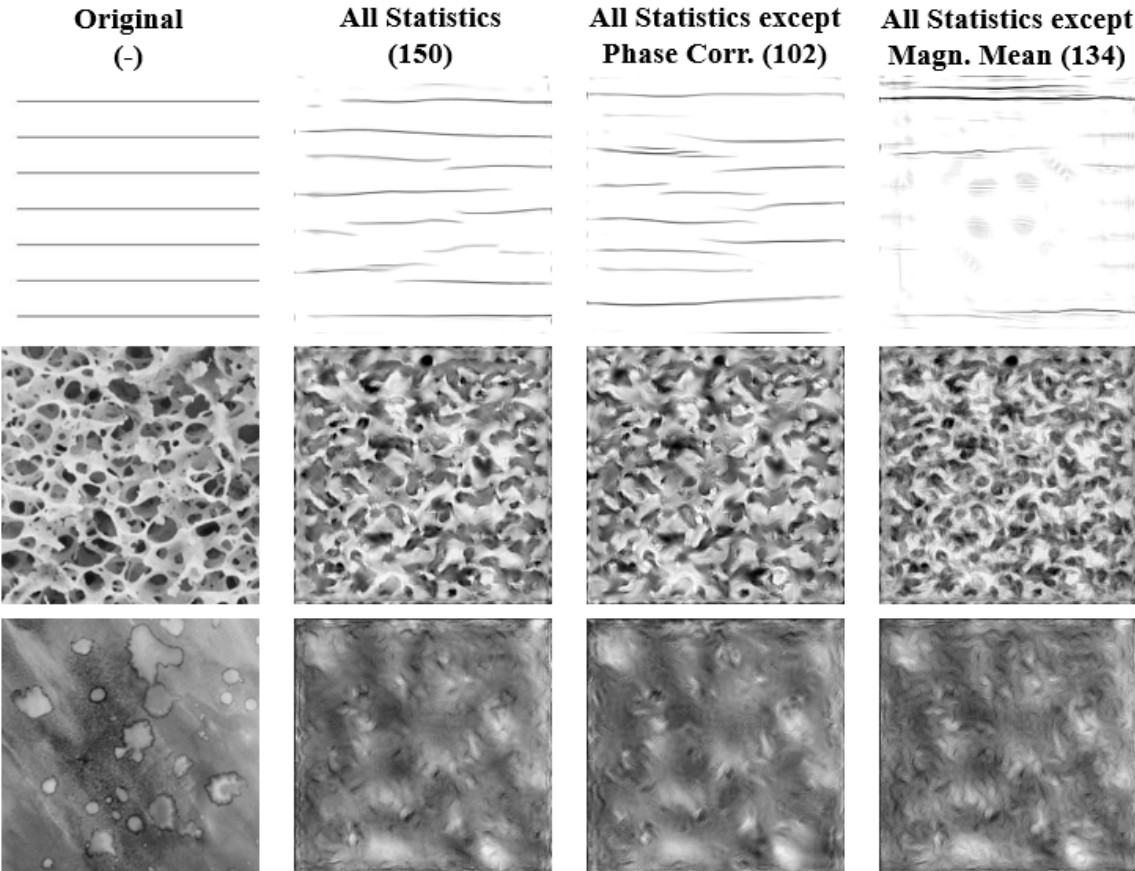

Figure 6: These three textures represent synthesis failures and success classes based on Brown et al. (2021). High contrast (first row, lined), middle contrast (middle row, porous) and low (bottom row, painted). Low roughness/coarseness textures (bottom row) have poor syntheses for even the full statistics set. Magnitude-mean is important for high and middle contrast textures as shown in the first two rows. Phase-correlation can be removed without much quality loss as compared to the full statistics synthesis.

## 10. Discussion

In this work, we combine self-supervised learning with weight interpretability analysis to develop, validate, and use a novel method that enables the principled selection and prioritization of the texture summary statistics underlying modern peripheral vision models. By adding a single fully-connected layer to a texture model, we create StatTexNet which we train with contrastive learning to prioritize the most important statistics on the task of grouping textures from the same family together. We show that StatTexNet successfully learns to group textures – indicating that it learned an optimal statistical representation of texture.

In addition, we use multiple weight interpretability metrics to order the relative contribution of individual statistics. To validate this ordering, we create a depleted texture set which is synthesized with a reduced set of statistics, train our network on these textures, and confirm that these reduced set of first-order statistics are the most important in grouping depleted textures as compared to original ones. We show that this result is consistent for 6 different orderings/metrics across 2 different datasets, averaged over multiple seeds.

Finally, we use this method to identify the relative importance of statistics in representing natural textures. When averaging over the sometimes heterogeneic texture families, we find that bandpass variance and magnitude-mean are the most important overall, while phase-correlation is least important. We show that our results are consistent not only with a small sample of synthesized textures, but also with previous psychophysical literature (Balas, 2006), which used psychophysical methodology to evaluate discrimination abilities for depleted textures. While their results found marginal statistics among the most important for the task of texture discrimination, like our work they find that cross-scale phase statistics to be among the least important for this task.

Overall, our method demonstrates a novel, efficient, and principled approach to selecting the statistics for peripheral vision models, as well as the pyramid-based texture-based models that underlie them. While a human in the loop will likely always be necessary to fully validate a statistics set, our method can make such experiments more directed, as testing all possible subsets of even 150 statistics in a formal eye-tracked psychophysics experiment is not feasible.

Future work could scale-up our approach using the larger set of statistics from models such as Portilla and Simoncelli (2000); Freeman and Simoncelli (2011); Rosenholtz et al. (2012), or a novel, much larger set of possible statistics. Additionally, the human visual system is thought to use highly complex transforms and performs a variety of tasks beyond grouping textures. Our method could be utilized to explore the effect of modeling more complex transformations on statistical importance, or the effect of alternative tasks such as classification, as more complex multi-layer weight structures are compatible with the Shapley method demonstrated here. Overall, with our principled and scalable approach to statistic selection, we can work toward better models of texture, peripheral vision, and human gaze as a whole.

## 11. Acknowledgements

This work was funded by the CSAIL MEnTorEd Opportunities in Research (METEOR) Fellowship, US National Science Foundation under grant number 1955219, as well as National Science Foundation Grant BCS-1826757 to PI Rosenholtz. The authors acknowledge the MIT SuperCloud Reuther et al. (2018) and Lincoln Laboratory Supercomputing Center for providing HPC resources that have contributed to the research results reported within this paper.

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

## 12. Appendix

### 12.1. Implementation

The implementation of this project is available as a Github repository at
https://github.com/RosenholtzLab/StatNetExperiments.

### 12.2. Contrastive Learning

Our aim is to develop a function $f_\theta(x) : \mathbb{R}^F \to \mathbb{R}^D$ that pushes encoded crops from the same classes in $\mathbb{R}^F$ closer together in $\mathbb{R}^D$. On the other hand, crops in $\mathbb{R}^F$ originating from different textures are pushed further apart in $\mathbb{R}^D$.

The function $f_\theta$ is parameterized by $\theta$. In this work $\theta$ represents the collective set of weights and biases of the neural network that are learned and adjusted during training to achieve the desired embeddings in the 50-dimensional space. For encoded textures $x$, the loss function employed is given through:

$$\mathrm{L}(\theta; X) = \sum_{i=1}^{P} \sum_{a=1}^{K} \left[ \log \left( \sum_{\substack{p=1 \\ p \neq a}}^{K} e^{D(f_\theta(x_a^i), f_\theta(x_p^i))} \right) + \log \left( \sum_{\substack{j=1 \\ j \neq i}}^{P} \sum_{n=1}^{K} e^{m - D(f_\theta(x_a^i), f_\theta(x_n^j))} \right) \right]_+$$

Here, the first term in the bracket are all positive pairs and the last term all negatives. The two summations indicate that we consider all pairs at once. As in (Hermans et al., 2017), the distance measure used is the Euclidean distance:

$$D(f\theta(x_i), f\theta(x_j)) = \|f\theta(x_i) - f\theta(x_j)\|_2$$

### 12.3. Data augmentation

For our self-supervised learning, we apply several transformations to the images. We use a random vertical flip with a 0.5 probability and a horizontal flip with the same probability. At the final step, we get five crops from the adjusted image: one from each corner and one from the center. These five crops all represent one class in the dataset and the contrastive learning setup. We avoided most transformations such as blurring or jittering because they could change the statistic values. After augmenting, we encode the five cropped images using the 150-statistic set. To keep the data consistent, we normalize the statistics with the Scikit standard scaler. This helps ensure our network is not influenced by varying statistic sizes. These normalized statistics are then processed through a single-layer network with input size 150 and output size of 50.

### 12.4. Labeling of statistics

The labeling of statistics is systematic, driven by their statistic group and the filter of the steerable pyramid they are derived from. We follow three distinct patterns of labeling.

- Non-correlation statistics: These are indicated in the format "statistic level orientation". For instance, "end stop 1 1" refers to the end stop statistic for the first orientation at the first pyramid level.

- Correlations between neighboring scales: This follows the format "statistic (level_1, level_2) orientation", i.e. "magnitude_correlation (2,3) 3", signifying a correlation between the second and third levels for the third orientation.

- Correlations within a level across different orientations: These are denoted as "statistic level (orientation_1, orientation_2)". This structure labels the correlation occurring within a specific pyramid level but across various orientations such as magnitude correlation 1 (1,3).

## 12.5. Correlations in Statistics

We expected that many of the statistics measured in our analysis were likely to be correlated due to the regularities present in natural images. To investigate the degree to which correlations between different statistics are present in our analysis, we calculated the correlation between each statistic over the dataset, then used Spearman Correlation to group the statistics.

We find that indeed, many statistics are highly correlated with each other. Marginals show strong correlation with other marginals, but little correlation with other statistics. The entire population of end-stopped and magnitude statistics together have strong correlation. In addition, there are strong repeated patterns of correlation and anti-correlation between phase and magnitude statistics. Statistics of the same type and scale/level share these patterns and cluster together.

## Clustered Spearman Statistics Correlation Heatmap

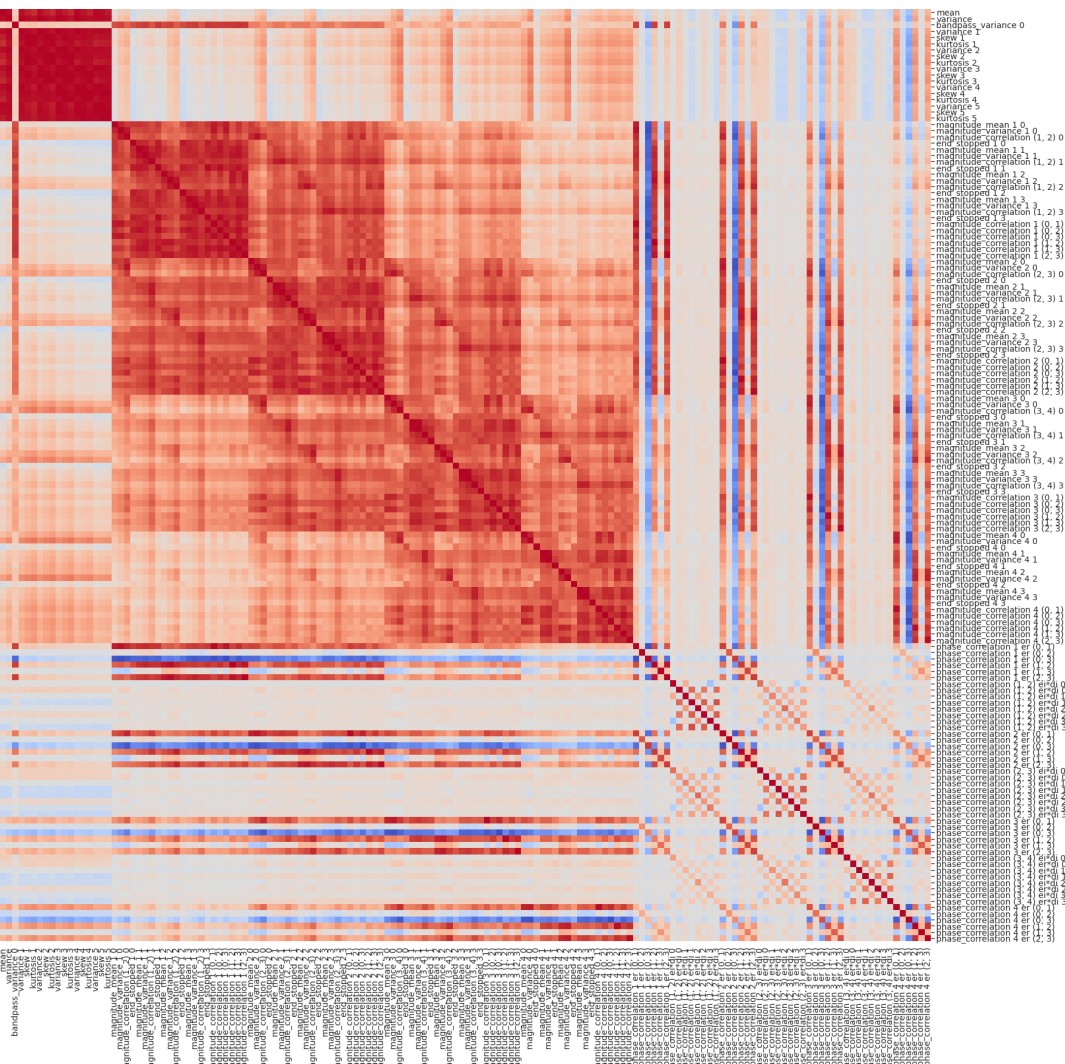

Figure 7: Correlation heatmap for all 150 statistics. Strong red color indicates positive correlation (1.0), while dark blue color anti-correlation (-1.0). There are high correlations between many statistics, especially within-group. There is also a subset of statistics that are anti-correlated or non-correlated.

## 12.6. Statistic Importance by Group

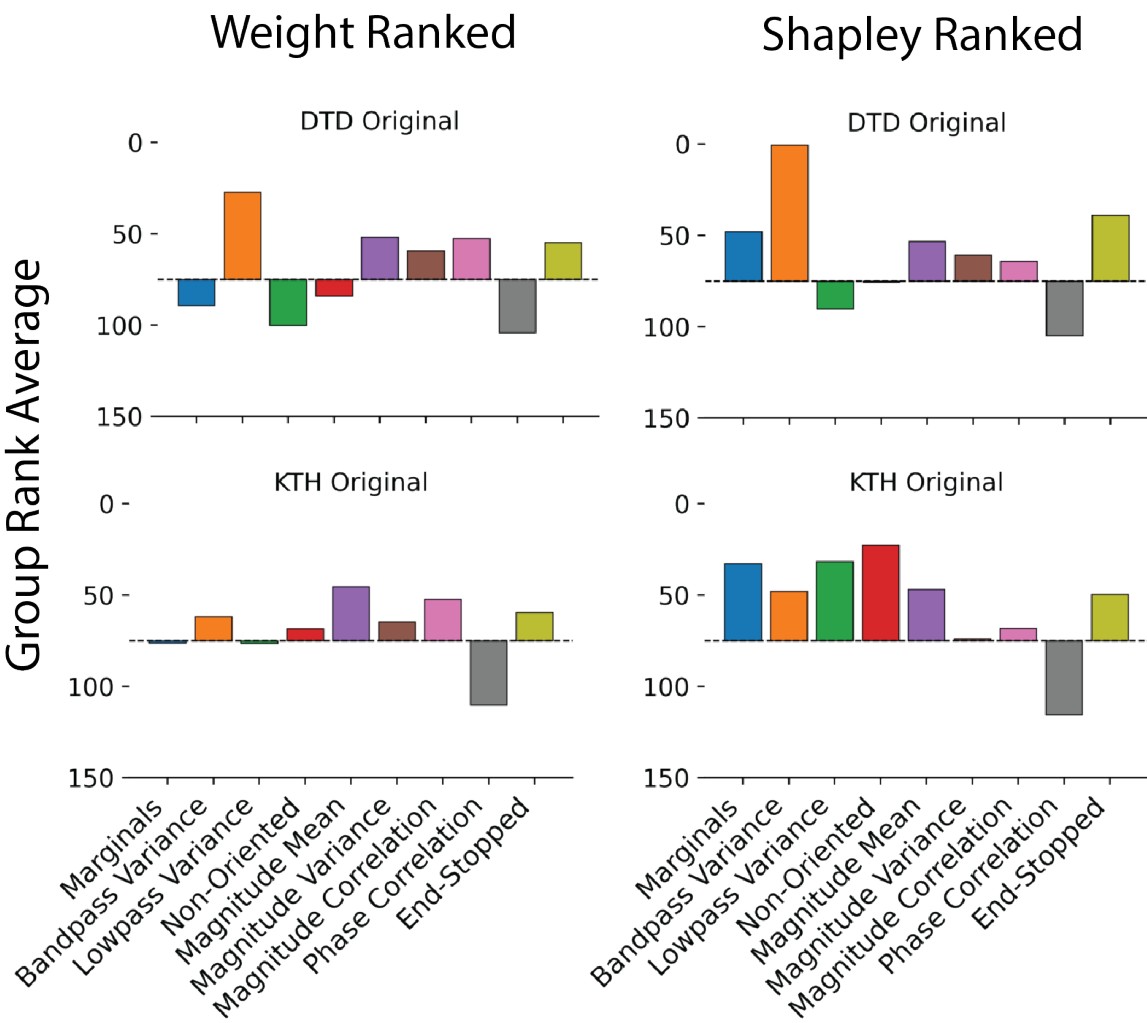

Figure 8: Mean ranking for the statistics groups with rankings based on weight (left), and Shapley values (right), for original textures of both datasets tested. Bandpass variance statistics generally rank with high importance, and phase-correlation statistics consistently rank with low importance.

## 12.7. Most and Least Important Statistics

Table 2: 10 most important statistics for the DTD & KTH dataset averaged over 10 seeds based on Shapley feature selection methods.

| DTD | | KTH | |
|---|---|---|---|
| Stat | Avg Rank | Stat | Avg Rank |
| end_stop 1 1 | 7.90 | end_stop 3 2 | 5.80 |
| end_stop 1 3 | 9.90 | magnitude_mean 3 3 | 13.70 |
| magnitude_correlation 1 (0, 2) | 10.50 | end_stop 3 0 | 14.30 |
| end_stop 1 0 | 11.40 | magnitude_mean 3 1 | 15.80 |
| end_stop 1 2 | 11.90 | magnitude_correlation (3, 4) 3 | 16.50 |
| magnitude_correlation 1 (1, 3) | 12.80 | magnitude_correlation 3 (0, 2) | 16.70 |
| magnitude_mean 1 3 | 14.20 | end_stop 2 2 | 17.80 |
| magnitude_mean 1 1 | 14.50 | magnitude_variance 3 3 | 19.00 |
| magnitude_mean 1 2 | 19.70 | magnitude_correlation (2, 3) 3 | 20.40 |
| magnitude_correlation 1 (0, 3) | 19.90 | magnitude_mean 4 3 | 20.50 |

Table 3: 10 least important statistics for the DTD & KTH dataset averaged over 10 seeds based on Shapley feature selection methods.

| DTD | | KTH | |
|---|---|---|---|
| Stat | Avg Rank | Stat | Avg Rank |
| phase_correlation (2, 3) er*di 1 | 132.30 | phase_correlation (2, 3) ei*di 1 | 135.90 |
| phase_correlation (2, 3) ei*di 2 | 133.10 | phase_correlation (1, 2) ei*di 1 | 136.10 |
| phase_correlation (2, 3) ei*di 0 | 133.50 | phase_correlation (1, 2) ei*di 3 | 136.10 |
| phase_correlation (2, 3) ei*di 1 | 133.50 | phase_correlation (3, 4) er*di 0 | 137.10 |
| phase_correlation 1 er (0, 2) | 134.30 | end_stop 4 2 | 137.90 |
| phase_correlation (2, 3) ei*di 3 | 135.10 | phase_correlation 1 er (0, 2) | 138.10 |
| phase_correlation 2 er (0, 2) | 136.20 | phase_correlation 2 er (0, 2) | 139.40 |
| end_stop 4 2 | 136.40 | phase_correlation (2, 3) ei*di 0 | 139.80 |
| phase_correlation (3, 4) ei*di 0 | 138.20 | phase_correlation (2, 3) er*di 0 | 140.00 |
| phase_correlation (3, 4) er*di 0 | 139.90 | phase_correlation (2, 3) er*di 2 | 140.60 |

