# OpenReview forum: "StatTexNet: Evaluating the Importance of Statistical Parameters for Pyramid-Based Texture and Peripheral Vision Models"
_NeurIPS.cc/2023/Workshop/Gaze_Meets_ML — Gaze Meets ML 2023 Poster_

### Official Review · Reviewer_Fa2c · 2023-10-20
**using contrastive learning to compress and select texture statistics for peripheral vision models**

**Rating:** 5
**Confidence:** 3

**Review:**

The paper aims to automate the process of statistics selection for textures extracted for peripheral vision models. Specifically, it uses a simple contrastive learning method to compress the statistics, and identifies which kind of statistics the network considers important using weights in the network and validates it using synthetic texture images created by considering the first-order statistics only. The paper is generally easy to follow. However, I have several questions regarding the method and its evaluations.
1. Is there any other machine learning based methods for texture statistics selection?
2. There is no comparison with other methods in the paper. Have the authors compared their method with other methods, for example, a simple autoencoder?
3. The feature representation is reduced from 150 to 50, why 1/3? how about other value?
4. When validating the method using synthetic/depleted texture images, was the model retrained using these depleted images? If so, how about those depleted images that are created by considering the second or third-order statistics? What would the order of weights be for these kinds of depleted images?

---

### Official Review · Reviewer_jUgW · 2023-10-21
**The authors propose a simple framework for textural data compression, and its effectiveness was evaluated with different methods.**

**Rating:** 6
**Confidence:** 3

**Review:**

The writing is clear in general, though some important details or rationales are missing. The network architecture itself is very simple, or too simple, as it only contains one single fully connected layer. On the other hand, the use of contrastive learning seems reasonable, and the use of multiple ways for evaluation is a merit. There are some questions regarding the choices of the hyperparameters:

1. Why only a single fully connected layer is used? Two such layers are usually used as two layers can approximate most functions in theory.
2. Why 50 output dimensions? Experiments that study how many output dimensions are required can be interesting.

---

### Official Review · Reviewer_tzAv · 2023-10-22
**Proposed a machine learning model (StatTexNet) for ordering the importance of the statistics in texture images**

**Rating:** 3
**Confidence:** 5

**Review:**

In this paper, the author proposed a deep learning-based methodology (called StatTexNet) to identify the importance of texture statistics. The authors use a pyramid-based network to get a reduced representation of the texture images and use contrastive learning to identify the significance of the statistics that differentiate the textures, Experiments ontwo datasets are used and different weighting metrics are used as significance metrics.

Main feedback:
1. It is not clear how this can be directly used for peripheral vision and gaze detection. In the reviewer's opinion, this paper probably is more suitable for some venues such as saliency detection in texture images
2.  The proposed paper lacks novelty. The technical details of the StatTxtNet are missing

---

### Meta-Review · Area_Chair_GW2G · 2023-10-26

**Recommendation:** Accept (Poster)
**Confidence:** 5

**Metareview:**

Peripheral vision is an important aspect of visual attention and common SOTA methods are pyramid-based texture models relying on manually selected statistics capturing text. The authors propose a novel methodology for selecting texture statistics automatically using self-supervised contrastive methods. The paper can be further improved if there is a simplified/accessible description of how statistics selection can improve peripheral vision understanding. Addressing most reviewers' comments should be trivial. Furthermore, comparison with other existing methods is much welcomed but not required.

---

### Decision · Program_Chairs · 2023-10-26

Accept (Poster)